

# Probing pair correlations in Fermi gases with Ramsey-Bragg interferometry

Théo Malas-Danzé[1,2], Alexandre Dugelay[1,2], Nir Navon[2,3] and Hadrien Kurkjian[4,5]

**1** ENS Paris-Saclay 91190 Gif-Sur-Yvette, France
**2** Department of Physics, Yale University, New Haven, Connecticut 06520, USA
**3** Yale Quantum Institute, Yale University, New Haven, Connecticut 06520, USA
**4** Laboratoire de Physique Théorique, Université de Toulouse,
CNRS, UPS, 31400, Toulouse, France
**5** Laboratoire de Physique Théorique de la Matière Condensée,
Sorbonne Université, CNRS, 75005, Paris, France

## Abstract

We propose an interferometric method to probe pair correlations in a gas of spin-1/2 fermions. The method consists of a Ramsey sequence where both spin states of the Fermi gas are set in a superposition of a state at rest and a state with a large recoil velocity. The two-body density matrix is extracted via the fluctuations of the transferred fraction to the recoiled state. In the pair-condensed phase, the off-diagonal long-range order is directly reflected in the asymptotic behavior of the interferometric signal for long interrogation times. The method also allows to probe the spatial structure of the condensed pairs: the interferometric signal is an oscillating function of the interrogation time in the Bardeen-Cooper-Schrieffer regime; it becomes an overdamped function in the molecular Bose-Einstein condensate regime.

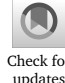
doi:[10.21468/SciPostPhys.17.1.024]

# 1 Introduction

At low temperatures, the behavior of quantum matter is often marked by the emergence of coherent ordered phases displaying remarkable macroscopic properties. Such condensed phases appear in various contexts, such as solid-state physics [1], nuclear or neutron matter [2], and ultracold atomic gases [3, 4]. They are characterized by long-range coherence carried by a macroscopically occupied wavefunction. In the simple case of the weakly interacting Bose gas, this order shows up as off-diagonal long-range order (ODLRO) in the one-body density matrix $\rho_1(\mathbf{r}, \mathbf{r}') = \langle \hat{\Psi}^\dagger(\mathbf{r}) \hat{\Psi}(\mathbf{r}') \rangle$ (where $\hat{\Psi}$ is the Bose field operator), such that $\lim_{|\mathbf{r}-\mathbf{r}'| \to \infty} \rho_1(\mathbf{r}, \mathbf{r}') = n_0$ is the density of the Bose-Einstein condensate (BEC). The ODLRO in a Bose gas has been measured for instance via the single-particle momentum distribution [5,6], which for a translationally invariant system is the Fourier transform of $\rho_1$.

In spin-1/2 Fermi systems, the one-body density matrix cannot exhibit ODLRO, owing to Pauli's exclusion principle, and the momentum distribution remains smooth across the superfluid phase transition [7]. Instead, a macroscopically occupied wavefunction signalling pair condensation can only appear in the two-body (pair) density matrix $\rho_2(\mathbf{r}_1, \mathbf{r}_2, \mathbf{r}'_1, \mathbf{r}'_2) = \langle \hat{\Psi}^\dagger_\uparrow(\mathbf{r}_1) \hat{\Psi}^\dagger_\downarrow(\mathbf{r}_2) \hat{\Psi}_\downarrow(\mathbf{r}'_2) \hat{\Psi}_\uparrow(\mathbf{r}'_1) \rangle$ (where $\hat{\Psi}_\sigma$ is the Fermi field operator for the fermion of spin $\sigma$) [3,8]. Measurements of ODLRO are for this reason considerably more challenging in Fermi systems. Rapid ramps of the magnetic field have been used to project the pair condensate onto a BEC of molecules [9–12]; however, the measured molecular condensed fraction is notoriously difficult to interpret theoretically, due to the various two- and many-body time scales involved in the problem [13]. Measurements of pair correlations in time-of-flight images have been proposed as a way to access ODLRO [14,15]; an analogous protocol has been implemented, albeit on a small Fermi system [16].

Interferometric protocols offer an alternative route to measure the coherence properties of quantum gases. Cold-atom experiments are particularly well suited for matter-wave interferometry, due to the possibilities of creating a coherent copy of the gas by manipulating the internal or external state of the atoms [17]. In Bose gases, direct real-space measurements of $\rho_1(\mathbf{r}, \mathbf{r}')$ were performed using Ramsey sequences based on interferometry of Bragg-diffracted gases [18–21]. In Fermi gases, matter-wave interference between small atom numbers extracted by spatially resolved Bragg pulses was proposed as a way to measure $\rho_2$ [22].

Inspired by such techniques, we propose a protocol to measure $\rho_2$ from the fluctuations of a Ramsey-Bragg interferometer. A copy of the spin-1/2 Fermi gas is created by imparting a large velocity to a fraction of the atoms. Interactions are turned off, and the copy travels ballistically, thereby stretching or translating the pairs of fermions by a distance proportional to the interrogation time. When the interferometric sequence is closed by the second pulse, the stretched and translated pairs interfere with those at rest, and a measurement of the correlations between the number of spin ↑ and spin ↓ recoiling atoms reveal the most important features of $\rho_2$. In the pair-condensed phase, the interferometric signal carries information on the magnitude of the fermionic condensate and on the wavefunction of the fermionic pairs.

# 2 Interferometric protocol

In Fig. 1 we show a sketch of the proposed measurement protocol. We consider a homogeneous spin-1/2 Fermi gas in a cubic box of size $L$ [23]. At $t = 0$, a first Bragg pulse is shined on the gas for a duration $t_{\text{pulse}}$. We place ourselves in the regime of a short and intense pulse, designed to be resonant with the whole gas and to create a moving copy of the cloud whose momentum distribution does not overlap with the original one (see Fig. 1). Both spin states are in a superposition of two components: a copy with no average momentum, and a copy with

a large average momentum $\mathbf{q}_{\mathrm{rec}}$. Assuming that the gas initially has zero mean velocity, the energy transferred by the pulse is adjusted to $\hbar\omega = \epsilon_{\mathbf{q}_{\mathrm{rec}}}$ (where $\epsilon_{\mathbf{k}} = \hbar^2 k^2/2m$ is the kinetic energy and $m$ is the mass of the fermion), in resonance with the atoms at rest. Since the atoms traveling at a velocity $\hbar\mathbf{k}/m \neq \mathbf{0}$ experience a detuning $\hbar\omega - \epsilon_{\mathbf{k}+\mathbf{q}_{\mathrm{rec}}} + \epsilon_{\mathbf{k}} = -\hbar^2\mathbf{q}_{\mathrm{rec}} \cdot \mathbf{k}/m$, the duration of the pulse $t_{\mathrm{pulse}}$ should be short enough so that this detuning remains negligible compared to the Fourier broadening over the typical range $\delta k$ of the momentum distribution of the gas:

$$\frac{\hbar q_{\mathrm{rec}} \delta k}{m} t_{\mathrm{pulse}} \ll 1. \tag{1}$$

Note that the pulse duration should also be long enough, *i.e.* $t_{\mathrm{pulse}} \gg m/\hbar q_{\mathrm{rec}}^2$, such that second-order transitions to states of momenta $\mathbf{k}+2\mathbf{q}_{\mathrm{rec}}$ or $\mathbf{k}-\mathbf{q}_{\mathrm{rec}}$ remain negligible. To evaluate the condition (1), let us consider the case of contact interactions between $\uparrow$ and $\downarrow$ fermions, characterized by an s-wave scattering length $a$. On the Bardeen-Cooper-Schrieffer side (BCS, $a < 0$), one can estimate $\delta k \approx \rho^{1/3}$, where $\rho$ is the total density, and on the molecular Bose-Einstein condensate side (BEC, $a > 0$) $\delta k \approx 1/a$. In this limit, the broadening of the momentum distribution implies that fulfilling $1/q_{\mathrm{rec}} \ll \hbar q_{\mathrm{rec}} t_{\mathrm{pulse}}/m \ll 1/\delta k$ will no longer be possible at fixed $q_{\mathrm{rec}}$.

In the intense-pulse regime of condition (1), the gas can be approximated by a two-level system undergoing Rabi oscillations between a state *at rest* (violet distribution in the upper sketches of Fig. 1) and a *recoiling* one (green distribution). The evolution during the first Bragg pulse corresponds to a rotation of angle $\theta = \Omega_R t_{\mathrm{pulse}}$ (where $\Omega_R$ is the Rabi frequency of the Bragg pulse) on the Bloch sphere of this effective two-level system:

$$\begin{pmatrix} \hat{a}_{\mathbf{k},\sigma} \\ \hat{a}_{\mathbf{k}+\mathbf{q}_{\mathrm{rec}},\sigma} \end{pmatrix}(t_{\mathrm{pulse}}) = \mathscr{S}(\theta, 0)\begin{pmatrix} \hat{a}_{\mathbf{k},\sigma} \\ \hat{a}_{\mathbf{k}+\mathbf{q}_{\mathrm{rec}},\sigma} \end{pmatrix}(0). \tag{2}$$

Here $\hat{a}_{\mathbf{k},\sigma}$ annihilates a fermion of wavevector $\mathbf{k}$ and spin $\sigma$ and the matrix $\mathscr{S}(\theta, \varphi) = \begin{pmatrix} \cos(\theta/2) & -i\sin(\theta/2)e^{i\varphi} \\ -i\sin(\theta/2)e^{-i\varphi} & \cos(\theta/2) \end{pmatrix}$ describes a rotation of angle $\theta$ around the vector $(\cos\varphi, -\sin\varphi, 0)$ of the equatorial plane of the Bloch sphere.

After this first pulse, the recoiling and non-recoiling components evolve ballistically during an interrogation time $\tau$. In contrast to the Ramsey-Bragg interferometry of weakly interacting gases [18, 20], it is crucial that interactions are turned off in strongly interacting gases before the first Bragg pulse. This would mitigate both fast many-body evolution during the interrogation sequence, and the high collisional density that would prevent the diffracted component from flying freely [24]. This could be achieved either with a fast Feshbach field ramp or with fast Raman pulses [16, 25]. The recoiling component travels a distance $\mathbf{x}_\tau \equiv \hbar\tau\mathbf{q}_{\mathrm{rec}}/m$, at a velocity sufficiently large to exit the trapping potential (in the direction of propagation). This means that only a fraction $(1 - x_\tau/L)$ of the cloud remains within the box volume after the interrogation time (assuming $\mathbf{q}_{\mathrm{rec}}$ is aligned with an axis of the cubic trap) and gives an upper limit $\tau < mL/\hbar q_{\mathrm{rec}}$ to the interrogation time.

After the interrogation time, the dephasing between the recoiling and non-recoiling components is $\varphi_{\mathbf{k}}(\tau) = ((\epsilon_{\mathbf{k}+\mathbf{q}_{\mathrm{rec}}} - \epsilon_{\mathbf{k}})/\hbar - \omega)\tau$ relatively to the Bragg transition, and a second Bragg pulse recombines the two components:

$$\begin{pmatrix} \hat{a}_{\mathbf{k},\sigma} \\ \hat{a}_{\mathbf{k}+\mathbf{q}_{\mathrm{rec}},\sigma} \end{pmatrix}(\tau + 2t_{\mathrm{pulse}}) = \mathscr{S}(\theta, \omega\tau)\begin{pmatrix} \hat{a}_{\mathbf{k},\sigma} \\ \hat{a}_{\mathbf{k}+\mathbf{q}_{\mathrm{rec}},\sigma} \end{pmatrix}(\tau + t_{\mathrm{pulse}})$$

$$= \mathscr{S}(\theta, \varphi_{\mathbf{k}}(\tau))\mathscr{S}(\theta, 0)\begin{pmatrix} \hat{a}_{\mathbf{k},\sigma} \\ \hat{a}_{\mathbf{k}+\mathbf{q}_{\mathrm{rec}},\sigma} \end{pmatrix}(0). \tag{3}$$

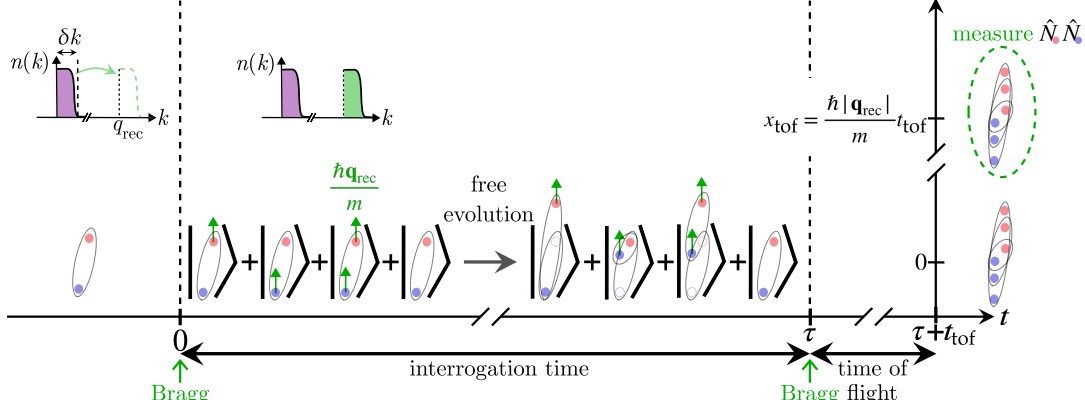

Figure 1: (a) Sketch of the Ramsey-Bragg interferometer applied to a pair of fermions. The blue (resp. red) circles represent spin ↑ (resp. ↓) atoms. The Bragg pulses create superpositions of atoms at rest and moving with a recoil momentum $q_{\text{rec}}$. After the time of flight, the component at rest and the recoiling one are separated by $x_{\text{tof}}$. For clarity, the finite pulse duration $t_{\text{pulse}}$ is not shown.

Eq. (3) thus describes a Ramsey sequence with a dephasing $\varphi_{\mathbf{k}}(\tau)$ that depends on the initial momentum of the atoms.[1] This makes the interferometer sensitive to the spatial structure of the gas, where short interrogation times allow to probe short-range correlations, and long times probing long-range correlations. Since the number of recoiling atoms is zero before the measurement sequence, the terms proportional to $\hat{a}_{\mathbf{k}+\mathbf{q}_{\text{rec}}}(0)$ can be omitted. For the operator describing the recoiling atoms at $t_{\text{f}} = \tau + 2t_{\text{pulse}}$ this gives

$$\hat{a}_{\mathbf{k}+\mathbf{q}_{\text{rec}},\sigma}(t_{\text{f}}) \to -\mathrm{i}\frac{\sin\theta}{2}\mathrm{e}^{-\mathrm{i}\epsilon_{\mathbf{k}+\mathbf{q}_{\text{rec}}}\tau}\left(1 + \mathrm{e}^{\mathrm{i}\varphi_{\mathbf{k}}(\tau)}\right)\hat{a}_{\mathbf{k}}(0). \tag{4}$$

After the Ramsey sequence, these recoiling atoms are spatially separated from the atoms at rest by a time of flight $t_{\text{tof}}$. An absorption image is taken to measure their number in each spin component:

$$\hat{N}_{\text{rec},\sigma} \equiv \sum_{\mathbf{k}\in\mathcal{B}}\hat{a}^{\dagger}_{\mathbf{k}+\mathbf{q}_{\text{rec}},\sigma}(t_{\text{f}})\hat{a}_{\mathbf{k}+\mathbf{q}_{\text{rec}},\sigma}(t_{\text{f}}) = \int \hat{\Psi}^{\dagger}_{\text{rec},\sigma}(\mathbf{r})\hat{\Psi}_{\text{rec},\sigma}(\mathbf{r})\,\mathrm{d}\mathbf{r}. \tag{5}$$

The summation over $\mathbf{k}$ is here restricted to the recoiling atoms, that is, to a neighborhood $\mathcal{B}$ of $\mathbf{q}_{\text{rec}}$ of typical size $\delta k$, small compared to $q_{\text{rec}}$. Using Eq. (4), we have expressed $\hat{N}_{\text{rec},\sigma}$ in terms of a field operator which superimposes atoms from different initial positions in the gas:

$$\hat{\Psi}_{\text{rec},\sigma}(\mathbf{r}) = \frac{\sin\theta}{2}\left(\hat{\Psi}_{\sigma}(\mathbf{r}) + \hat{\Psi}_{\sigma}(\mathbf{r}-\mathbf{x}_{\tau})\right), \tag{6}$$

where $\hat{\Psi}_{\sigma}(\mathbf{r}) = (1/\sqrt{L^3})\sum_{\mathbf{k}\in\mathcal{B}}\mathrm{e}^{-\mathrm{i}\mathbf{k}\cdot\mathbf{r}}\hat{a}_{\mathbf{k},\sigma}(0)$ is the field operator at $t=0$. Consequently, pairs of recoiling atoms are described by the pairing field $\hat{\Psi}_{\text{rec},\downarrow}\hat{\Psi}_{\text{rec},\uparrow}$, which yields the superposition depicted in Fig. 1:

$$\hat{\Psi}_{\text{rec},\downarrow}(\mathbf{r}_2)\hat{\Psi}_{\text{rec},\uparrow}(\mathbf{r}_1) = \frac{\sin^2\theta}{4}\big[\hat{\Psi}_{\downarrow}(\mathbf{r}_2)\hat{\Psi}_{\uparrow}(\mathbf{r}_1) + \hat{\Psi}_{\downarrow}(\mathbf{r}_2)\hat{\Psi}_{\uparrow}(\mathbf{r}_1-\mathbf{x}_{\tau})$$
$$+ \hat{\Psi}_{\downarrow}(\mathbf{r}_2-\mathbf{x}_{\tau})\hat{\Psi}_{\uparrow}(\mathbf{r}_1) + \hat{\Psi}_{\downarrow}(\mathbf{r}_2-\mathbf{x}_{\tau})\hat{\Psi}_{\uparrow}(\mathbf{r}_1-\mathbf{x}_{\tau})\big]. \tag{7}$$

The four terms here represent respectively a pair at rest, a pair where the ↑ or the ↓ fermion has been stretched by $\mathbf{x}_{\tau}$, and a pair globally translated by $\mathbf{x}_{\tau}$.

---

[1] Note that the dephasing $\varphi_{\mathbf{k}}(2t_{\text{pulse}})$ accumulated during the two Bragg pulses is negligible by virtue of Eq. (1).

## 3 Measuring long-range pair ordering

As in Bose gases, the measurements of $\hat{N}_{\text{rec}}$ give access to one-body correlations:

$$\hat{N}_{\text{rec},\sigma} = \frac{\sin^2\theta}{2}\big[\hat{N}_\sigma + \hat{\rho}_{1,\sigma}(\mathbf{x}_\tau)\big], \tag{8}$$

where $\hat{\rho}_{1,\sigma}(\mathbf{x}_\tau) = \int \hat{\Psi}_\sigma^\dagger(\mathbf{r})\hat{\Psi}_\sigma(\mathbf{r} - \mathbf{x}_\tau)\mathrm{d}\mathbf{r}$ is the one-body correlation operator and $\hat{N}_\sigma$ is the total number of atoms of spin $\sigma$; we assumed that $\hat{\rho}_{1,\sigma}$ is symmetric under parity, *i.e.* $\hat{\rho}_{1,\sigma}(-\mathbf{x}_\tau) = \hat{\rho}_{1,\sigma}(\mathbf{x}_\tau)$.

In Fermi gases, $\rho_2$ is more interesting since it is the observable that exhibits long-range (pair) order. To measure $\rho_2$, we propose to record the correlations of the numbers of spin ↑ and ↓ recoiling atoms:

$$S(\mathbf{x}_\tau) = \big\langle \hat{N}_{\text{rec},\uparrow}(\mathbf{x}_\tau)\hat{N}_{\text{rec},\downarrow}(\mathbf{x}_\tau)\big\rangle - \big\langle \hat{N}_{\text{rec},\uparrow}(\mathbf{x}_\tau)\big\rangle\big\langle \hat{N}_{\text{rec},\downarrow}(\mathbf{x}_\tau)\big\rangle. \tag{9}$$

Such interferometric signal is constructed by averaging individual realizations of $N_{\text{rec},\uparrow}$ and $N_{\text{rec},\downarrow}$. Using Eq. (7) to express the quartic part of $S$, we recognize the following contractions of $\rho_2$:

$$f_{\text{tr}}(\mathbf{x}_\tau) = \int \rho_2(\mathbf{r}_1 - \mathbf{x}_\tau, \mathbf{r}_2 - \mathbf{x}_\tau; \mathbf{r}_1, \mathbf{r}_2)\,\mathrm{d}\mathbf{r}_1\mathrm{d}\mathbf{r}_2\,, \tag{10}$$

$$f_{\text{str},\uparrow}(\mathbf{x}_\tau) = \int \rho_2(\mathbf{r}_1 - \mathbf{x}_\tau, \mathbf{r}_2; \mathbf{r}_1, \mathbf{r}_2)\,\mathrm{d}\mathbf{r}_1\mathrm{d}\mathbf{r}_2\,, \tag{11}$$

$$f_{\text{str},\downarrow}(\mathbf{x}_\tau) = \int \rho_2(\mathbf{r}_1, \mathbf{r}_2 - \mathbf{x}_\tau; \mathbf{r}_1, \mathbf{r}_2)\,\mathrm{d}\mathbf{r}_1\mathrm{d}\mathbf{r}_2\,, \tag{12}$$

$$f_{\text{str},\uparrow\downarrow}(\mathbf{x}_\tau) = \int \rho_2(\mathbf{r}_1 - \mathbf{x}_\tau, \mathbf{r}_2; \mathbf{r}_1, \mathbf{r}_2 - \mathbf{x}_\tau)\,\mathrm{d}\mathbf{r}_1\mathrm{d}\mathbf{r}_2\,. \tag{13}$$

These functions have a simple interpretation: $f_{\text{tr}}$ measures the overlap between the translated and the original pair of Eq. (7), $f_{\text{str},\sigma}$ the overlap between the pair stretched by the spin $\sigma$ fermion and the original one, and $f_{\text{str},\uparrow\downarrow}$ the overlap between the two pairs stretched by the fermion of the opposite spin. Using Eq. (8) for the quadratic part of $S$, we finally obtain:

$$S = \frac{\sin^4\theta}{4}\left[ f_{\text{str},\uparrow} + f_{\text{str},\downarrow} + \frac{f_{\text{str},\uparrow\downarrow} + f_{\text{tr}}}{2} - \rho_{1,\uparrow}\rho_{1,\downarrow} - N_\uparrow\rho_{1,\downarrow} - N_\downarrow\rho_{1,\uparrow} \right], \tag{14}$$

where $\rho_{1,\sigma} \equiv \langle\hat{\rho}_{1,\sigma}(\mathbf{x}_\tau)\rangle$. The signal $S$ is maximum for $\theta = \pi/2$; we thus set $\theta$ at this value from now on. When the gas is in the normal phase, the functions $f_{\text{str}}$, $f_{\text{tr}}$ and $\rho_1$ vanish at large distances. On the contrary, when the gas is pair condensed, the contribution of the translated pairs $f_{\text{tr}}$ does not vanish in the thermodynamic limit when $x_\tau \to +\infty$. In this case, $\rho_2$ has a macroscopic eigenvalue $N_0$ associated to a wavefunction $\phi_0$ and behaves at large distances (that is, when the pair centers of mass $\mathbf{R} = |\mathbf{r}_1 + \mathbf{r}_2|/2$ and $\mathbf{R}' = |\mathbf{r}_1' + \mathbf{r}_2'|/2$ are infinitely separated) as

$$\lim_{|\mathbf{R}-\mathbf{R}'|\to+\infty} \rho_2(\mathbf{r}_1, \mathbf{r}_2, \mathbf{r}_1', \mathbf{r}_2') = N_0\phi_0^*(\mathbf{r}_1, \mathbf{r}_2)\phi_0(\mathbf{r}_1', \mathbf{r}_2'). \tag{15}$$

This implies that $\lim_{x_\tau\to+\infty} f_{\text{tr}}(\mathbf{x}_\tau) = N_0$, such that

$$S_\infty \equiv \lim_{x_\tau\to+\infty} S(\mathbf{x}_\tau) = \frac{N_0}{8}\,. \tag{16}$$

We have assumed here that fluctuations of the total atom numbers, if there are any, are uncorrelated: $\langle\hat{N}_\uparrow\hat{N}_\downarrow\rangle = N_\uparrow N_\downarrow$. Eq. (16) provides a direct measurement of the magnitude $N_0$ of the

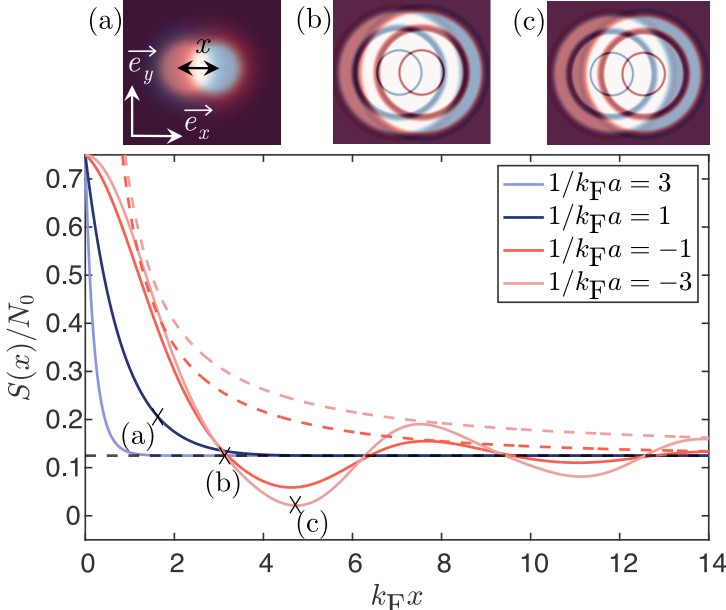

Figure 2: The interferometric signal $S(x)$ as a function of the distance $x$ for different values of the interaction strength, calculated using the mean-field BCS theory (solid curves); here, we assume $x = x_{\tau,\uparrow} = x_{\tau,\downarrow}$. On the BCS side, where $S$ oscillates, the envelope is $(x_0/\pi x)\exp(-x/\xi_x)$ (dashed lines). (a)-(c) Sketches of the interference patterns for $S$ originating from the condensate wavefunction $\phi_0$. The copy at rest is shown in blue ($|\phi_0(\mathbf{r}_1, \mathbf{r}_2)|^2$) and the translated one in red ($|\phi_0(\mathbf{r}_1, \mathbf{r}_2 + \mathbf{x}_\tau)|^2$), where $x = |\mathbf{x}_\tau|$; (a) in the BEC regime, (b) in the BCS regime, where the displacement $x$ corresponds to the first cancellation of $S$ (see main panel), and (c) in the BCS regime, where the displacement corresponds to the first minimum of $S$.

long-range order, a quantity that cannot be measured using the rapid ramp technique [9,10]. Note that $N_0$ cannot be interpreted as the number of condensed pairs away from the BEC limit.[2]

The contribution of the stretched pairs to $S$ through $f_{\text{str},\sigma}$ and $f_{\text{str},\uparrow\downarrow}$, although negligible at distances greater than the pair size $\xi_{\text{pair}}$, carries essential information on the condensate wavefunction $\phi_0$. It is possible to isolate the contribution of $f_{\text{str},\sigma}$ using a spin-selective Bragg pulse, such that the displacements $\mathbf{x}_{\tau,\uparrow}$ and $\mathbf{x}_{\tau,\downarrow}$ of the two spins no longer coincide. For $\mathbf{x}_{\tau,\downarrow} = \mathbf{0}$ and $\mathbf{x}_{\tau,\uparrow} \neq \mathbf{0}$, Eq. (14) becomes

$$S(\mathbf{x}_{\tau\uparrow}) = \frac{f_{\text{str},\uparrow}(\mathbf{x}_{\tau\uparrow}) - N_\downarrow \rho_{1,\uparrow}(\mathbf{x}_{\tau\uparrow})}{2}. \tag{17}$$

This result can be used to reveal the momentum structure of $\phi_0$. Let us suppose that the system is isotropic and translationally invariant. If the pairs are tightly bound (as in the BEC limit), then $\phi_0(\mathbf{r}_1, \mathbf{r}_2)$ decreases rapidly and almost monotonically with $x = |\mathbf{r}_1 - \mathbf{r}_2|$, and so does $f_{\text{str},\sigma}$; the corresponding behavior for $S$ is schematically depicted in Fig. 2(a). Conversely, if pairing occurs at a non-zero wavenumber, as in the BCS limit, $\phi_0$ oscillates as a function of $x$ at a wavelength corresponding to the inverse of that wavenumber, and so does $f_{\text{str},\sigma}$ (see Figs. 2(b)-(c)).

---

[2] The pair-condensate annihilation operator $\hat{b}_0 = \int \phi_0^*(\mathbf{r}_1, \mathbf{r}_2)\hat{\Psi}_\downarrow(\mathbf{r}_1)\hat{\Psi}_\uparrow(\mathbf{r}_2)\mathrm{d}\mathbf{r}_1\mathrm{d}\mathbf{r}_2$ is not bosonic, as $\langle[\hat{b}_0, \hat{b}_0^\dagger]\rangle \leq 1$ (the inequality is saturated only in the BEC limit). Therefore, $N_0 = \langle\hat{b}_0^\dagger\hat{b}_0\rangle$ is not the number of pairs in the condensate in the general case.

 Select                                          SciPost Phys. 17, 024 (2024)

# 4 BCS mean-field approximation

To obtain a more explicit expression for $S$, and illustrate its behavior when $x_\tau \approx \xi_{\text{pair}}$, we now use the BCS mean-field approximation and assume that the gas is balanced, such that $N_\uparrow = N_\downarrow$, $f_{\text{str},\uparrow} = f_{\text{str},\downarrow}$ and $\rho_{1,\uparrow} = \rho_{1,\downarrow}$. The total density $\rho = \rho_\uparrow + \rho_\downarrow$ defines the Fermi wavenumber $k_F = (3\pi^2\rho)^{1/3}$, and in the BCS state $\rho_2$ factorizes into

$$\rho_2(\mathbf{r}_1, \mathbf{r}_2, \mathbf{r}'_1, \mathbf{r}'_2) = N_0 \phi_0^*(\mathbf{r}_1, \mathbf{r}_2)\phi_0(\mathbf{r}'_1, \mathbf{r}'_2) + \rho_1(\mathbf{r}_1, \mathbf{r}'_1)\rho_1(\mathbf{r}_2, \mathbf{r}'_2). \tag{18}$$

If the gas is translationally invariant and isotropic, the functions previously defined in Eqs. (10)-(13) depend only on $x_\tau = |\mathbf{x}_\tau|$. Since symmetry-breaking BCS states do not have a fixed number of particles, there is a nonzero covariance $\langle \psi_{\text{BCS}}|\hat{N}_\uparrow \hat{N}_\downarrow|\psi_{\text{BCS}}\rangle \neq N_\uparrow N_\downarrow$. We get rid of this well-known artifact of BCS theory by projecting the BCS states onto the subspace with a fixed number of atoms (see *e.g.* Eq. (41) in [26]). The interferometric signal in the case $\mathbf{x}_{\tau,\uparrow} = \mathbf{x}_{\tau,\downarrow}$ [Eq. (14)] becomes:

$$S(x_\tau) = \frac{N_0}{8}\Big[1 + 4f(x_\tau) + f(2x_\tau)\Big]. \tag{19}$$

Here the function

$$f(x) = \int \phi_0^*(\mathbf{r}_1 - \mathbf{x}, \mathbf{r}_2)\phi_0(\mathbf{r}_1, \mathbf{r}_2)\mathrm{d}\mathbf{r}_1\mathrm{d}\mathbf{r}_2, \tag{20}$$

is the overlap between a stretch and an original pair of the condensate; it is related to the functions introduced before by $f_{\text{str},\sigma} = N_0 f + N_\sigma \rho_1$ and $f_{\text{str},\uparrow\downarrow}(x) = N_0 f(2x) + \rho_1^2(x)$. The condensate wavefunction in Fourier space $\phi_\mathbf{k}$, defined by $\phi_0(\mathbf{r}_1, \mathbf{r}_2) = \sum_\mathbf{k} \phi_\mathbf{k} e^{-i\mathbf{k}\cdot(\mathbf{r}_1 - \mathbf{r}_2)}/L^3$, takes the form

$$\phi_\mathbf{k} = \frac{\Delta}{2E_\mathbf{k}\sqrt{N_0^{\text{BCS}}}}, \tag{21}$$

where $\Delta$ is the gap, $E_\mathbf{k} = \sqrt{(\epsilon_\mathbf{k} - \mu)^2 + \Delta^2}$ is the BCS dispersion relation, and $\mu$ is the chemical potential. The associated macroscopic eigenvalue is $N_0^{\text{BCS}} = \sum_\mathbf{k} \Delta^2/(4E_\mathbf{k}^2)$. The maximum of $|\phi_\mathbf{k}|$ is reached at the minimum of the BCS dispersion relation, that is, at $k_{\min} = \sqrt{2m\mu}/\hbar$ on the BCS side ($\mu > 0$) and $k = 0$ on the BEC side ($\mu < 0$). Using the BCS condensate wavefunction Eq. (21), we can calculate the integral over $\mathbf{k}$ analytically in Eq. (20), which yields

$$f(x) = e^{-x/\xi_x}\text{sinc}(\pi x/x_0), \tag{22}$$

where the exponential decay length

$$\xi_x^2 = \frac{\hbar^2}{m\Delta}\left(\frac{\mu}{\Delta} + \sqrt{1 + \frac{\mu^2}{\Delta^2}}\right), \tag{23}$$

can be identified with the characteristic length of the one-body density matrix [27, 28], and

$$\frac{x_0^2}{\pi^2} = \frac{\hbar^2}{m\Delta}\frac{1}{\frac{\mu}{\Delta} + \sqrt{1 + \frac{\mu^2}{\Delta^2}}}, \tag{24}$$

is the oscillation length.

Oscillations of $S$ are visible before $S$ reaches its asymptotic value depending on the ratio $x_0/\xi_x$. In the BCS limit ($\mu/\Delta \to +\infty$ or $k_F a \to 0^-$), the oscillation length $x_0 \sim \pi/k_F$ is much shorter than the exponential-decay length $\xi_x \sim \hbar^2 k_F/m\Delta$ which diverges as $O(\xi_{\text{pair}})$. Thus, in the BCS regime, $S$ exhibits oscillations (the dark and light red curves in Fig. 2 correspond to $1/k_F a = -1$ and $-3$); the oscillations decay as a cardinal sine, on a typical length scale $1/k_F$.

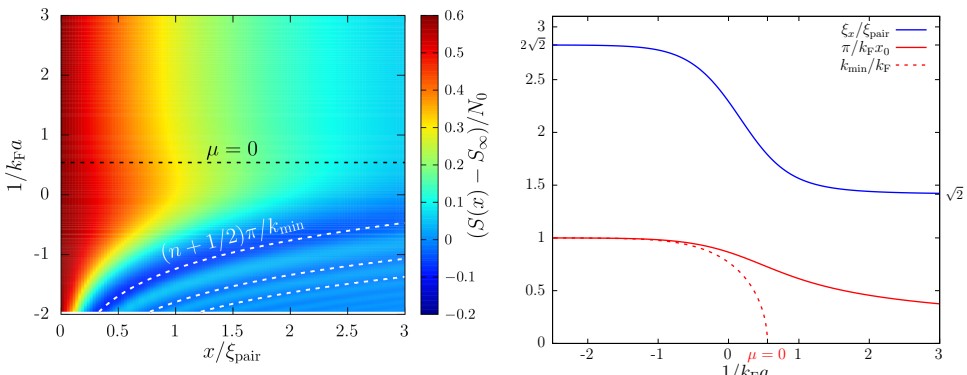

Figure 3: (Left panel) The interferometric signal $S(x) - S_\infty$ normalized to $N_0$ as a function of $x/\xi_{\mathrm{pair}}$ and $1/k_{\mathrm{F}}a$ within the mean-field BCS approximation. The boundary between the BEC and BCS regime ($\mu = 0$ at $1/k_{\mathrm{F}}a \simeq 0.54$) is marked by the black dashed line. On the BCS side, we compare the local minima of the oscillatory signal to $x_n = (n + 1/2)\pi/k_{\mathrm{min}}$ (white dashed curves). (Right panel) The wavenumber $\pi/x_0$ (normalized to $k_{\mathrm{F}}$) and the exponential attenuation length $\xi_x$ (normalized to the Cooper pair size $\xi_{\mathrm{pair}}$) of the overlap function $f$ in the BEC-BCS crossover. The dashed red curve shows the location of the dispersion minimum $k_{\mathrm{min}} = \sqrt{2m\mu}/\hbar$ on the BCS side ($\mu > 0$).

Conversely, in the BEC limit ($\mu/\Delta \to -\infty$ or $k_{\mathrm{F}}a \to 0^+$), $\xi_x \sim a$ tends to zero like the size of the bosonic dimers. At the same time, the oscillation frequency diverges as $x_0 \sim \sqrt{3\pi/4k_{\mathrm{F}}a}\,(\pi/k_{\mathrm{F}})$, such that no oscillations are visible in this regime (the dark and light blue curves on Fig. 2 correspond to $1/k_{\mathrm{F}}a = 1$ and 3). A transition between the two regimes (illustrated in the left panel of Fig. 3) occurs around the point where $\xi_x = x_0/\pi$, that is, $\mu = 0$, which coincides with the point where the minimum $k_{\mathrm{min}}$ of the BCS dispersion relation reaches 0. A measurement of the BCS gap is also accessible through the relation

$$\frac{\xi_x x_0}{\pi} = \frac{\hbar^2}{m\Delta}. \tag{25}$$

In Fig. 3, we compare $\xi_x$ to the pair size defined as [29]

$$\xi_{\mathrm{pair}} = \left( \int \rho_2(\mathbf{r}_1, \mathbf{r}_2, \mathbf{r}_1, \mathbf{r}_2) |\mathbf{r}_1 - \mathbf{r}_2|^2 \mathrm{d}\mathbf{r}_1 \mathrm{d}\mathbf{r}_2 \Big/ \int \rho_2(\mathbf{r}_1, \mathbf{r}_2, \mathbf{r}_1, \mathbf{r}_2) \mathrm{d}\mathbf{r}_1 \mathrm{d}\mathbf{r}_2 \right)^{1/2} \tag{26}$$

(see the blue line), showing that the two quantities remain comparable throughout the BEC-BCS crossover.[3] We also compare the wavenumber $\pi/x_0$ of the overlap function $f$ to the location of the dispersion minimum $k_{\mathrm{min}} = \sqrt{2m\mu}/\hbar$: they coincide in the BCS limit but differ outside, in particular because $\pi/x_0$ does not vanish (solid red curve on Fig. 3), unlike $k_{\mathrm{min}}$ (dashed red line).

While our quantitative discussion of $S(x)$ is restricted to the mean-field approximation, we note that $\rho_2$ in general, and the contractions introduced in (10)–(13) in particular, have been computed using more advanced diagrammatic approximations [27]. Away from the BCS

---

[3]We derived the analytic expression:

$$\xi_{\mathrm{pair}}^2 = \frac{\hbar^2}{2m\Delta} \frac{4\alpha^2(\alpha + r_\alpha) + 7\alpha + 5r_\alpha}{8r_\alpha(\alpha + r_\alpha)},$$

where $\alpha = \mu/\Delta$ and $r_\alpha = \sqrt{1 + \alpha^2}$.

limit, where fluctuations in the bosonic collective modes become important, a slower decay than the exponential one predicted by Eq. (22) is expected, which is reminiscent of the power-law convergence of $\rho_1$ to the condensed fraction in a Bose gas [30].

In summary, we proposed an interferometric protocol to probe the two-body density matrix in spin-1/2 Fermi gases. By measuring the correlations between the recoiling atoms of $\uparrow$ and $\downarrow$ after a Ramsey-Bragg sequence, one records as a function of the interrogation time a damped oscillatory signal whose attenuation time, frequency, and asymptotic value give access all at once to the size of the Cooper pairs, to their relative wave number, and to the macroscopic eigenvalue of the two-body density matrix. Those important features of fermionic condensates are difficult to access experimentally [31]. Furthermore, this method has the advantage that a fine spatial resolution on $\rho_2$ is obtained through a fine temporal resolution, which is rather easy to achieve experimentally. The correlation signal recorded at the end of the sequence also involves a macroscopic fraction of the atoms initially present in the trap, which makes it more robust to experimental noise. In the future, it would be interesting to extend this calculation to the case of fermions with three internal states [32].

# Acknowledgments

We thank S. Huang and G. Assumpção for insightful discussions. H.K. thanks Yale University for its hospitality.

**Funding information**   This work was supported by the NSF (Grant Nos. PHY-1945324 and PHY-2110303), DARPA (Grant No. HR00112320038), AFOSR (Grant No. FA9550-23-1-0605), the EUR grant NanoX n° ANR-17-EURE-0009 in the framework of the "Programme des Investissements d'Avenir". N.N. acknowledges support from the David and Lucile Packard Foundation, and the Alfred P. Sloan Foundation.

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
