# Peer review of "Probing pair correlations in Fermi gases with Ramsey-Bragg interferometry"

_SciPost Physics, doi:SciPost Phys. 17, 024 (2024)_

## Round 3 · Referee Report · Anonymous (Referee 1) · 2024-4-28

Report

The authors have addressed all points and suggestions raised in my previous referee report. I am fully in favor of its publication on Scipost Physics.

Recommendation

Publish (easily meets expectations and criteria for this Journal; among top 50%)

---

## Round 3 · Referee Report · Anonymous (Referee 2) · 2024-5-17

Report

In their manuscript, Malas-Danze et al. introduce an innovative experimental protocol aimed at measuring key characteristics of the two-body density matrix in a spin-1/2 Fermi system. This protocol leverages Ramsey-Bragg interferometry to create a superposition of the spin-up and spin-down components from both the initial zero-momentum cloud and a high-average-momentum cloud. By examining the correlations between the recoiling spin-up and spin-down atoms, the interferometric signal uncovers detailed information about the presence of off-diagonal long-range order (ODLRO) associated with a pair condensate, as well as the internal spatial structure of the pairs. These aspects are difficult to probe experimentally, making this protocol a significant advancement in the field.
The authors tackle a critical problem in this field and offer a highly promising method backed by robust theoretical analysis. The experimental design is detailed and methodologically sound, showcasing the protocol’s potential to drive progress in the field. The ability to directly demonstrate superfluid ODLRO in an ultracold Fermi gas would be a significant achievement. This method’s precision in measuring such elusive features is particularly noteworthy.
Overall, the manuscript is well-written and logically structured, effectively conveying the significance and potential impact of the research. Given the novelty and the rigor of the approach, as well as its implications for advancing our understanding of fermionic condensates, I recommend that this paper be published in SciPost Physics. The proposed protocol not only addresses a challenge in the field but also opens new avenues for experimental exploration and theoretical development.

Recommendation

Publish (easily meets expectations and criteria for this Journal; among top 50%)

---

## Round 3 · Author Response

Warnings issued while processing user-supplied markup:

  • Inconsistency: Markdown and reStructuredText syntaxes are mixed. Markdown will be used.
    Add "#coerce:reST" or "#coerce:plain" as the first line of your text to force reStructuredText or no markup.
    You may also contact the helpdesk if the formatting is incorrect and you are unable to edit your text.

Dear Editors of SciPost,

We thank you for considering our manuscript, and the reviewer for their careful reading of our manuscript. Below, we address the reviewer’s comments, point by point. We made the corresponding changes in the manuscript, as listed in the list of changes.

The authors

=== Response to the reviewer ===

1) Can the authors explain better how the duration of the pulse appears in inequality (1)?

The method that we propose requires that the Bragg-Ramsey pulses have the same pulse area for all classes of momenta in the gas. This inequality guarantees that the Fourier broadening is large enough so that all such classes of momenta see the same pulse area. We have adjusted the wording to make the explanation clearer. Also, incidentally, we found that there were incorrect factors of hbar in Eq. (1) and below, which we corrected.

2) In the second to the last sentence of the paragraph following eq (1), it is stated “fulfilling both inequalities”. However, only one inequality is introduced before this sentence (the second inequality is introduced in the last sentence of the paragraph).

We thank the reviewer for this relevant comment. We actually noticed this after the submission and corrected it on the version we uploaded on the arXiv shortly after. The text after Eq. (1) has been modified (but still in black).

3) I found the notation \Psi_{r,sigma}(r) for the recoil field operator rather confusing. It took me some time to grasp that the first “r” stand for recoil, the second for the position r. Could the authors introduce an alternative notation? For instance, using \psi (lowercase) for the recoil operator, or \Psi_{rec}?

That was indeed a potential source of confusion. We changed all the recoil ‘r’ for ‘rec’.

4) The definition for the recoil operator in Eq. (4) is crucial. However, it comes with little explanation (a few words are given after Eq. (4)). Can the authors provide more details?

We clarified this definition by substantially rewriting this section of the manuscript. We now introduce first the number of recoiling atoms (which is simply a restriction of the total number to momenta around q_{rec}), and explain that this operator can be expressed in terms of a field operator (introduced by the new Eq. (6)) combining atoms for different positions of the gas. To make the derivation clear, we now give Eq. (4) as an intermediate step of the calculation.

5) I found a bit confusing that Eq. (13) for S does not contain the term <\hat{N}\up \hat{N}_down> - N_up N_down, but then this term pops up in Eq. (18). The reason is that <\hat{N}\up \hat{N}down> - N_up N_down = 0 if the number of particles is conserved, while this term is different from zero in Eq. (18) because the BCS wave-function does not conserve the number of particles. I think that the reader would be helped if a further line would be added in eq. (13) where the term <\hat{N}\up \hat{N}down> - N_up N_down is still present, and then in the next line this term is dropped (and the reason for dropping it would be explained immediately after). 7) The term <\hat{N}\up \hat{N}_down> - N_up N_down, which is different from zero within BCS, would become zero after projection of the BCS wave-function onto a number conserving wave-function. I thus wonder if it would not be better to drop altogether its contribution to S(x) (with appropriate explanation), in particular when presenting Fig. 2. We know that this constant term is spurious. In this way S(x) of Fig. 2 would become closer to the real physical S(x).

We thank the reviewer for these very useful complementary remarks. It was indeed confusing to have the covariance of the total numbers in the BCS result but not in the general one. We have thus removed the covariance from Eq. (19) (in the BCS case) by seizing the reviewer’s suggestion to project the BCS states onto the subspace with fixed total numbers N_\up and N_\down.

6) The function S(x) in Fig. 2 approaches the ODLRO exponentially, over a length scale \xi_x which is related to the pair size. However, one would expect on general grounds that due to the presence of the Goldstone gapless mode in the broken symmetry phase, the asymptotic limit (yielding the condensate fraction) should be reached as a power law (see e.g. the book by D. Forster “Hydrodynamic Fluctuations, Broken Symmetry, and correlation functions (1975)”). The point is that to take into account this mode one should go beyond BCS mean field when calculating \rho_2 and include pairing fluctuations. This has been recently done by L. Pisani et al., PRB 105, 054505 (2022). The function h_2 shown in Fig. 9 of that paper corresponds to f_tr -1/2\rho_{1, up}\rho_{2,down} of the present manuscript. Within mean-field, this function would be constant, and equal to the condensate density. The inclusion of a Maki-Thompson-like diagram makes this function to depend on R and approach its asymptotic value as a power law. The various terms f_str of the present work, on the other hand, which are related to the stretching of a pair, should instead be described reasonably well by BCS mean-field. In summary, with the inclusion of fluctuations, the function S(x) shown in Fig. 2 should display first an exponential behavior over a scale \xi_x, followed by a power law decay when the contribution coming from f_str fades away, and only f_str survives. I suggest the authors to comment on this in their manuscript.

This was another very insightful point. We followed the reviewer’s suggestion, modified Fig. 2 accordingly and added a paragraph after Eq. (18).

8) Eq. (22) for the length scale \xi_x coincides with the T-> 0 limit of Eq. (A11) for the length scale \xi_1 in the above work by Pisani et al. In that reference, \xi_1 is the scale over which the one-body density matrix decays. In addition, the same result is obtained by J.C. Obeso et al., New J. Phys 25, 113019 (2023) (see their eqs. (14) and (15)) for the scale \xi_alpha over which density-density correlators decay. Can the authors add a comment on this in their manuscript?

We thank the reviewer for pointing out these useful references, we now mention them directly after Eq. (23).

---

## Round 3 · List of Changes

Changes compared to the previous version, by order of apperance in the text:

1- The sentence "Note that the pulse duration should also be long enough \hk{\emph{i.e.} $ \tpulse \gg m/{\hbar q_{\rm {rec}}^2}$} such that second-order transitions to states of momenta $\vk+2\vqr$ or $\vk-\vqr$ remain negligible." has been moved to just after equation (1), in accordance with the text. A factor $\hbar$ has also been corrected.

2- The paragraph beginning with "After the Ramsey sequence, the recoiling atoms...", including eqs 5 and 6, has been substancially rewritten to clarify the derivation of the number of recoiling atoms. We also added the new equation (4) to make the progression easier to follow.

3- We have redrawn fig 1 to improve its description of the time-evolution of the gas.

4- We have removed the covariance from eq 19, as we now assume that our mean-field state is the superposition of BCS states that has a fixed number of atoms. The asymptotic value of Fig 2 has been modified, and the former footnote 3 removed accordingly.

5- We have added Ref 27 and 28 below eq 23

6- We have added the paragraph "While our quantitative discussion..." to discuss beyond mean-field effects

---

## Editorial Decision

published